# "I want to recycle batteries, but it's inconvenient": A Study of Non-rechargeable Battery Recycle Practices and Challenges

Category: Research

### ABSTRACT

Non-rechargeable batteries are widely used in electronic devices and can cause environmental issues if not recycled properly. However, little is known about the challenges that people might encounter when they recycle non-rechargeable batteries. We first conducted an online survey with 106 participants to understand their practices and challenges of reusing and recycling non-rechargeable batteries. We then interviewed 12 participants to understand the potential reasons behind their behaviors. Our results show that although it is common to store used batteries temporarily, many eventually do not recycle them for reasons such as the inconvenience of recycling, not knowing how to recycle batteries and high perceived efforts of recycling. Moreover, we highlight the challenges associated with their common battery reuse and recycle strategies. We present design considerations and potential solutions for both individuals and communities to promote sustainable battery recycle behaviors.

**Index Terms:** Human-centered computing—Human-computer interaction—Empirical studies in HCI;

## 1 INTRODUCTION

Non-rechargeable batteries, also known as primary batteries, are commonly used in portable electronic devices (e.g., remote controls, stereo headsets) and are expected to grow at a compound annual growth rate of around 3% to nearly $19 billion by 2022 [2]. Americans purchased nearly 3 billion primary batteries yearly [1].

Although recycling batteries is beneficial to the environment [3] and is encouraged by governments (e.g., [1]), only 36% of used batteries were estimated to be collected and 29% were recycled in the European Union in 2015 [1]. Collectively, each person in the US discards 8 primary batteries per year [1]. By 2025, approximately 1 million metric tons of spent battery waste will be accumulated [12]. However, little is known about how people recycle used batteries and what the potential challenges are. To fill in this gap, We sought to answer the following two research questions(RQs):

- RQ1: What are the practices and challenges of reusing and recycling used batteries?

- RQ2: What are design opportunities to improve using and recycling used batteries?

We first conducted a survey study with 106 participants living in North America to understand their practices and challenges of reusing and recycling used batteries. Informed by the results, we further conducted in-depth interviews to explore people's willingness and barriers to conducting environmental-friendly approaches towards dealing with used batteries and major factors that affect participants' decision-making of reusing and recycling batteries.

Our results show that although it is common to store used batteries temporarily, many eventually do not recycle them for reasons such as the inconvenience of recycling, lack of information about recycling batteries, and high perceived efforts of recycling batteries. Moreover, the practices of reusing batteries depend on the financial status, the motivation to conduct environmentally-friendly behavior, and the availability of tools (e.g., battery testers) and instructions. To our knowledge, this is the first study that provides both quantitative and qualitative understanding of battery reuse and recycle practices and challenges from users' perspectives.

## 2 BACKGROUND AND RELATED WORK

### 2.1 Regulations of Collecting Batteries

Previous research showed that people should avoid the simple behavior of discarding household batteries along with municipal solid waste because the collection, separation, and recycling processes are accessible worldwide [4]. For example, Japan, United States, and European countries have equipped the whole countries with official recycling programs with collection centers. End users are the first part of the collection chain who must return spent batteries, while the second ones are distributors and manufacturers who should perform their responsibilities to collect batteries free of charge. Therefore, we decided to explore the individual behaviors of dealing with batteries and how the community or government helped individuals involve in environmental-friendly activities regarding battery recycle.

A study [15] found that the impacts on the environment of the collection activities, closely bounded with the transportation, outbalanced the benefit to the environment. To minimize the negative impact of transportation, several countries in Europe applied the method of "integrated waste management" to integrate the collection of batteries and other recyclable material. Take Sweden as an example, battery [15]. So do the trucks that transported both paper and batteries. A project in the Netherlands would extract old batteries from household waste with magnets [3]. We also aimed to study combined with the living environment and people's behavior, how we can practice "integrated waste management" to reduce the side effect generated from battery recycling activities.

### 2.2 Individual Battery Recycle Behavior

Researchers investigated people's environmental attitudes and opinions and found that there is a positive relationship between general environmental attitudes and recycling actions though it is a fairly tenuous relationship [8, 11, 14, 18]. Several research studies suggested that to increase the citizens' participation in recycling, it is useful to educate the public about the significance of recycling and to inform them of how and where to recycle [16, 17]. Previous research suggested the major reasons why people refuse to recycle used batteries are due to the cost of time and effort and a lack of material reward [13, 19]. To be more specific, consumers are more willing to recycle objects when it is convenient to access and use the recycling equipment [20]. Moreover, when the disposal method is integrated into everyday life, individuals feel encouraged to take actions that they assume are sustainable [19]. According to interviews with families with children in the Netherlands [7], most families expressed their unsatisfactory that recycle bins are not always available when they recycle household items, like glass or plastic bottles. Xiao et al. [7] conducted a survey that explored certain generation's actions and thoughts of recycling e-waste, as well as the barriers to recycling. The results show that individuals' practices vary largely. In the analysis process, they classified the recycling actions into five categories, including transferring a product to other users, returning it to the manufacturer, and reusing the object. Accordingly, we aimed to further investigate the major reasons and barriers of reusing and recycling batteries in everyday life for North-America residents. This would provide design implications for human-computer interaction researchers to best design tools and methods to assist people with reusing and recycling batteries.

## 3 SURVEY

### 3.1 Survey Design

The survey included 14 multiple-choice, Likert-scale, and short-answer questions, which were organized into themes to elicit data about participants' practices and opinions about the devices with non-rechargeable batteries that they use, reusing batteries, and recycling batteries as well as their knowledge of non-rechargeable batteries regulations which differ from place to place.

### 3.2 Procedure and Participants

We distributed the survey via email lists from a university and social media platforms, such as Facebook and Slack, between March and November 2020. We received 107 responses, removed one duplicate response, and performed the analyses on the 106 valid responses.

79% of the participants (N=84) were from the USA and 21% (N=22) were from other countries. 54 participants were between 18 and 25, 42 were between 26 and 35, 7 were between 36 and 50, and 4 were above 50.

### 3.3 Findings

#### 3.3.1 Reusing Batteries

Participants were presented with a scenario that "the TV remote control uses three single-use batteries, and you find that the batteries cannot provide enough power." and provided their inferences on the batteries' conditions and also their potential solutions.

While about a third (32%) of the participants believed that all the batteries were *completely* drained, the majority (67%) of them believed that some of the batteries were only *partially* drained. Nonetheless, only 52% chose to *keep some of the batteries for later reuse*, and 41% chose to *change the batteries with new ones all at once*. This highlights a gap between participants' understanding of the used batteries and their potential actions to deal with them.

One major challenge of reusing batteries is to find out how much power is left in a used battery. However, 74% of the participants reported having *no or little experience with testing the remaining power of a used battery*. Only 2 participants (less than 2%) reported having such experience.

#### 3.3.2 Recycling Batteries

Participants were asked to report whether and how they might recycle used batteries. 77% (N=82) of the participants chose to *"store the used batteries temporarily"*, 32% (N=32) chose to *"throw the used batteries into a regular trash can"*, and only 14% (N=15) chose to *"take the used batteries to a recycling center"*. The most frequently-mentioned barriers of visiting a recycling center were as follows: *I do not know where to recycle the batteries* (N=69), *I do not collect many batteries* (N=48), it is inconvenient to visit a recycling center(N=43), and *I do not have incentives to do so* (N=19).

Furthermore, there were challenges for recycling batteries. 70% of the participants *did not know the regulations and laws of recycling batteries in their local area*. Only 22% *sought the resources and information about recycling batteries*.

## 4 INTERVIEWS

To further understand the challenges of reusing and recycling used batteries and identify opportunities to improve reusing and recycling practices, we conducted a semi-structured interview study.

### 4.1 Interview Design

The interview is consists of 4 parts. In part 1, we provided the descriptions about the difference between non-rechargeable batteries and rechargeable batteries to avoid confusion about the concepts. In addition, we asked a kick-off question about the recent experience of using batteries to prepare participants for exploring the problems that they encountered in daily life. Part 2 focused on people's practice and knowledge under two typical scenarios to learn their practice and willingness of replacing and reusing batteries. Part 3 covered previous experience in recycling batteries. Besides, we asked about the experience with other recyclable objectives and looked for good and bad reference of providing convenience to personal recycle activities. Part 4 unveiled our idea that people can donate or receive old batteries from others to make full use of the batteries. We sought participants' opinions on this idea and discovered elements that affects their decision-making.

### 4.2 Participants

We recruited 12 participants from the survey respondents, social media platforms, and word-of-mouth. 4 participants were identified as males and 7 as females; 7 participants were 18-24 years old, 4 were 25-35 years old, and 1 was 36-50 years old. 11 participants lived in the US and 1 in Canada. 5 participants had recycling experience in more than 1 country and 1 participant had related experience in 2 states in the US. Each participant was compensated with $10.

### 4.3 Procedure

The study obtained approval to conduct the interview from the Institutional Review Board of Rochester Institution of Technology. We conducted the study with participants remotely with an online meeting platform, such as Zoom, Google meeting. The interview session lasted for about 30 40 minutes. The whole interview sessions were audio-recorded using the Voice Memos application and transcribed the interview content using Otter.ai.

### 4.4 Analysis

Two authors first performed open coding and discussed about disagreements on coding to gain a consensus. They then performed an affinity diagramming to derive themes emerging from codes.

### 4.5 Findings

Our analysis revealed the rationales and challenges associated with the practices of reusing and recycling batteries as well as the potential design opportunities.

#### 4.5.1 Reusing Batteries

*Battery usage behaviors vary depending on people's tolerance of the perceived interruptions when products run out of power.* Participants tend to change all batteries at once for products that would cause high perceived disruptions to their user experience when running out of power, for example, the controller of a video game console. In contrast, they would be more willing to change only one of the batteries for products that would cause low perceived interruptions when running out of power, such as a TV remote controller.

Our survey results show that when the batteries cannot serve a product, 67% of the respondents believe that the batteries are only partly used. In the interview, we investigated whether they are willing to measure the voltage in the battery.

Only two out of the 12 participants indicated that they had battery testers to measure the remaining power or voltage in the batteries and decide whether they would reuse the batteries. All other participants showed little interest in knowing the leftover power in the batteries and indicated that they would replace all batteries together. We found three reasons. First, it was perceived to be *time-consuming* to test with new batteries and replace all the old batteries one by one. Secondly, they did not *feel the need to save batteries* in particular when did not have many devices using single-use batteries.

#### 4.5.2 Recycling Batteries

70% of the survey respondents were not confident about their knowledge about battery recycling regulations. The interview study further explored people's willingness to learn the related regulations.

*Willingness to learn about regulations:* Seven out of the twelve participants indicated that they would like to learn the regulations about recycling batteries; two participants did not care much about the regulations; three would not actively seek to learn the regulations but would learn when encountering them. P10 mentioned that *"I don't actively seek it on my own initiative, but if I accidentally see it, I would click in."*

There is certain content that people are interested in. 7 of 12 participants want to learn where they could discard or recycle batteries. Except for that, law, specific rules and regulations, and knowledge about processing the batteries.

8 out of 12 participants were aware that they should recycle non-rechargeable batteries or they could not discard these batteries in the regular trash. However, only 1 participant knew where to discard the non-rechargeable batteries and regulation in where he/she lived. Only 2 participants had or have experience in recycling non-rechargeable batteries. 4 participants indicated that they knew where to discard the non-rechargeable battery in other countries or districts, including China, Canada, Taiwan, and Turkey.

**Challenge.** In the survey, only 16% of participants had experience in visiting recycle places to recycle batteries. In the interview, we tried to figure out the reason why they did not go to the recycle places (willingness) and in which way they considered it an easy way to recycle (challenge).

All the participants in the interview attempted to dispose of rechargeable batteries in the right place now or before to some extent. P5 said that *" I know that they should be recycled in some way but I don't know where. So I threw it in the trash."* However, only P8 had the habit of recycling batteries because his working space had a disposing location.

P4 used to recycle batteries but felt it was hard to recycle batteries after she moved to a new place 6 years ago, and she mentioned her emotions: *"Once we (our family) were very distressed about disposing of the battery, but we didn't do deep search on the Internet. I feel this is a very simple thing, but so hard to find one. ... We used to live in a small town. There was a university near where we lived, and there were battery recycling places in it. "*

Two participants indicated that it was inconvenient to recycle batteries because they did not use a lot of batteries because *it would be too much work*. The other two participants thought that recycling batteries was part of the state laws and regulations. Although P11 was unfamiliar with the regulations and laws, he felt it was reasonable for residents to recycle batteries because *"this is how we move the societies forward by being strict on these environmentally friendly things that aren't too difficult to do."* P3 felt that the current law was not strict enough to regulate residents' behavior and people are less likely to care about it. He mentioned that *"People are not just willing to push this into law so they are less likely to care about this. ...I don't care as much because it's not a law. "*

### 4.5.3 Design Opportunities

Our interviews also revealed three design opportunities.

**Making recycling convenient to people**. Locations, where participants and their family members visited to recycle batteries, include convenience stores, universities, apartment leasing office, electronic retailer stores, and their workplaces. One common characteristic of these locations was that they were all convenient for participants to visit. In particular, when they had to run an errand near these recycle locations, they would be more willing to visit the location.

Participants proposed several places to position battery recycling facilities (e.g., a bin): 1) places near where people live, for example, next to regular trash drop-off locations in a residential community; 2)locations near or in the stores that people visit regularly, for example, grocery stores, wholesales stores, or convenience stores; 3) libraries: P4 and P12 both mentioned that libraries are places that

families with children and students often visit, and 4) recreational centers where people do sports and attend recreational classes.

**Learning from practices of recycling other items**. We asked participants about other items that they recycled in their daily lives and why they were able to recycle them. The frequently recycled items included paper and newspaper, cardboard, and plastic bottles and cans. The main reason why these items got recycled often was that participants could simply put these items next to their regular trash bins and wait for waste management to collect them. This finding shows again that convenience is key to recycling. Moreover, small rewards (e.g., store credits) were given by certain grocery stores to encourage people to recycle plastic bottles and cans. However, P5 felt that rewards might not work for recycling batteries because the tedious process of collecting and bringing batteries to certain locations as well as the hygienic issues associated with used batteries outweigh the small rewards grocery stores could provide.

**Making information about how and where to recycle batteries easy to access**. 65% of the survey participants did not know where they should send batteries to. Thus, we investigated the reasons in the interviews. Results show that participants sought out battery recycle regulations and locations primarily from their social circles, such as their parents, spouses, friends, and landlords. Surprisingly, few participants used searching online (e.g., Google) as a way to find out battery recycle regulations and locations in their local areas. *"Because we cannot find it via the internet, at least we tried to google...but we couldn't find it."*-P5

Participants proposed six approaches to delivering battery recycle information that would be convenient for them to spot: 1) on the *packages of the devices that use batteries*, which show information about how to recycle batteries or a QR code that can be easily scanned by a smartphone; 2) on the *battery brands' websites*. An important consideration is to make sure such websites would pop up on the top of the search results list; 3)*local governments*: Local governments could inform their residences of relevant information via text messages, emails, news reports, or bulletins; 4) *landlords, housing agents or dorm managers*: They could be helpful for people who recently moved to the area and are not familiar with local battery recycle regulations and guidelines; 5) *non-profit organizations*: non-profit organizations could use public promotion activity and educational videos to show people the alarming consequences of discarding batteries without properly recycling them; and 6) *waste management companies*: waste management companies could also help residents recycle batteries, such as setting up a hot-line.

The preferred formats to deliver battery recycling regulations and guidelines were *info-graphics, short videos, social media posts, or advertisements*. Info-graphics could be displayed on a battery's packaging or on the packaging of a product that uses batteries.

## 5 Discussion

Informed by the findings of both the survey and interview studies, we present design considerations (DCs) for designers and researchers to consider when helping people better reuse and recycle batteries.

**DC1: Help Users Understand the State of Used Batteries and How They Could Reuse them**. Our studies found that two prominent challenges of *reusing batteries* were: 1) *it was unknown whether a battery was fully drained or there was some power left*; 2) *what other products they could reuse the batteries for*. Reasons for these challenges included *lacking tools and knowledge about how to test the power of a used battery* and *having no easy access to information about possible products that could take used batteries*.

To help implement this design consideration, we propose the conceptual designs of a portable battery tester and its companion mobile app to illustrate how to lower the barrier for the general public to test used batteries and find information about how to reuse and recycle them. Figure 1 (a) and (b) show two views of the battery tester that contains three slots to place three common types

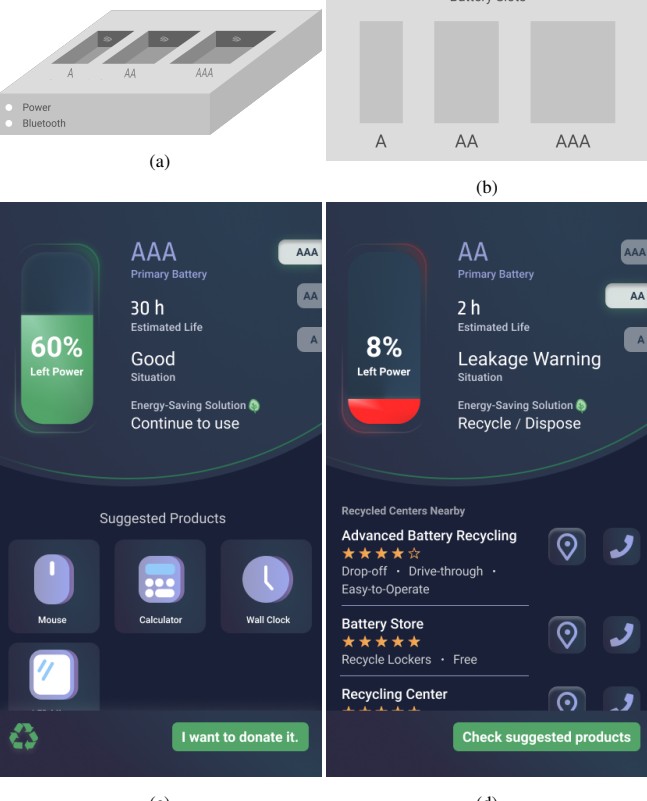

Figure 1: Battery Tester Prototype and companion Mobile App: (a) A side view of the prototype; (b) a top view of it; (c) App UI that shows that the tested AAA battery is in a good condition and offers a list of products that it could be reused in; (d) App UI that shows the tested AA battery was ready to be disposed or recycled and offers information about how it could be properly handled.

of non-rechargeable batters. The battery tester could connect with a smartphone via Bluetooth and send the test results to be displayed in the mobile companion app. Figure 1 (c) and (d) show the UIs of the app, which show the battery test results and the recommended actions for two batteries in good and bad conditions respectively. Making all of the information needed for reusing and recycling batteries in one app would save users efforts and time to perform random searches online, which were reported to be less effective by our participants and also prior studies [16, 17].

**DC2: Make recycling batteries integrated into people's routine lives**. Our studies show that inconvenience was one key barrier to recycling batteries. This finding corroborates the findings that showed that the cost of time and effort and lack of material rewards hinder people from recycling batteries [13,19]. Our participants who did recycle batteries often had relatively easy access to recycling locations, such as their workplace, nearby stores, universities, and the community and homeowner's associations. This finding echos the suggestion of previous research [19].

To help implement this design consideration, we recommend HCI designers and researchers consider the successful practices of recycling other materials, such as cardboard, cans, and plastic bottles, and make recycling batteries embedded in people's everyday lives [9]. For example, waste management companies would take the recyclable materials weekly along with the trash and the community would set a separate space next to the regular trash for residents to drop off recyclable materials. Furthermore, some grocery stores

have recycle facilities for people to recycle bottles and cans and even offer small rewards. These potential example solutions align with a concept of "integrated waste management" [3] and should be considered when integrating the collection of batteries with other waste streams to minimize not only individual's recycling efforts but also the negative impact associated with the transportation of batteries.

**DC3: Build community supports to help people recycle batteries**. Our studies also found that for those who did manage to recycle batteries, they received some community supports. For example, one interviewee mentioned that her previous community manager would notify the residents to carry used batteries to the community office once or twice a year.

To help implement this design consideration, we propose to build online community platforms for people to share information about and also exchange used batteries. A successful example platform for people to exchange used items is Cragslist [6]. There are several challenges to overcome. First, it remains unclear how to motivate people to participate in such platforms. One approach might be to gamify the process to make it fun and rewarding to participate. Second, unlike other used items, used batteries may cause hazardous concerns if not handled properly during the sharing process. Similar to our proposed design in Figure 1, it is worth designing simple approaches to checking batteries' conditions.

## 6 LIMITATIONS AND FUTURE WORK

First, this short paper focused on understanding the practices and challenges of reusing and recycling batteries and deriving design considerations. Although we also offered potential solutions (e.g., Figure 1), they are yet to be fully implemented and evaluated with users. Second, the individual's practice of recycling batteries may vary [7]. In our interviews, we also noticed differences between participants who lived alone and those who lived with their families or roommates. More research is needed to investigate how battery reuse and recycle practices might be affected by social factors, such as the number of people that they live with. Lastly, our analysis highlights the importance of informing individuals of recycling locations and times in their local areas. However, our participants often had difficulty in finding such information. Although many online resources (e.g., earth911 [10] and call2recycle [5]) provide such information, future research should further investigate the barriers in users' information searching process and design tools to help users easily find such information.

## 7 CONCLUSION

We have conducted a survey study and an interview study to understand the practices and challenges of reusing and recycling batteries. Our results found various barriers in the way of reusing and recycling batteries. First, due to the lack of information, people do not know the efficient ways to recycle batteries. Even though some may have the good intention of conducting environmental-friendly practices, people are unwilling to invest too much time and effort if they could easily access necessary information. Regarding reusing or making full use of batteries, people have difficulties in figuring out the remaining power of batteries efficiently. As a result, many would discard batteries that still have power left, which leads to the waste of energy. Our analysis also uncovered opportunities to lower the barriers to reuse and recycle batteries. Finally, we present three design considerations and discuss potential solutions.

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
