# OpenReview forum: ""I want to recycle batteries, but it's inconvenient": A Study of Non-rechargeable Battery Recycle Practices and Challenges"
_graphicsinterface.org/Graphics_Interface/2021/Conference/Second_Cycle — Reject_

### Official Review · Reviewer_EB8v · 2021-04-22
**This paper presents the results of a online survey and interview study of North American participants, about non-rechargeable battery recycling practices and barriers.**

**Rating:** 3
**Confidence:** 5

**Review:**

This paper has several significant weaknesses, which led to this reviewer's decision for rejection. Some suggestions on moving forward are below.

1. The overall topic and research questions that this paper explores were not framed in a compelling way.
- First, the introduction lacked a justification for why the authors chose to investigate NON-rechargeable batteries (since rechargeable batteries are easily accessible in the marketplace and already commonly used).
- Second, the barriers and challenges to recycling non-rechargeable batteries seem like an issue to be addressed by city infrastructure, sustainability policies and incentives -- rather than a problem to be addressed by technology design. (Also see related comment #4 below).

2. The Related work section: The authors discuss recycling infrastructure in European countries like Sweden and the Netherlands, but this study is about North American cities, where recycling infrastructure differs greatly (even between cities). Thus, the motivation for conducting this work was weakly motivated. The literature on recycling practices/barriers in North American cities should also have been discussed.

3. Consequently, from #2, the online study and the accompanying interview study were not well motivated. Methodology was lacking detail on what questions or themes were explored, and most importantly, why the authors asked these questions. This reviewer was also unclear why the interview study was needed, after conducting the web study? This justification was missing.

4. Section 4.5.3 (design opportunities) -- Related to comment #1 above, none of the design opportunities were tech-oriented. Rather, many of these must be addressed by city recycling infrastructure.

5. Section 5 (discussion) is a bit strange, as it is not a discussion of the findings but rather proposes more "design considerations" (DC). DC1 makes a potentially interesting point but it is strange that suddenly a prototype appears in the discussion section.  DC2 and DC3 are not at all related to technology. DC3 talks about Craigslist as a design opportunity to build a community to support recycling practices but this is not written in a compelling way.

6. Finally, throughout the paper, there a lot of grammar mistakes and sentence fragments. This is most apparent in Section 2.1, 2.2 and 4.3, and others are throughout.

*** Suggestions for moving forward:
- Ask whether this study is asking the right questions, or whether the focus should be on something else. Are the questions you are asking meaningful? Important or significant to the tech community? Are they too broad?
- Consider scoping this study to explore how to motivate people to re-use non-rechargeable batteries with other devices (e.g. as mentioned in 3.3.1). However -- again, perhaps this is not the most interesting or significant question to investigate.
- If motivation is a key variable the authors wish to explore, there is tons of related work on this that the authors could incorporate to motivate or inspire their work.  Eg.

Alsalemi, A., Sardianos, C., Bensaali, F., Varlamis, I., Amira, A., & Dimitrakopoulos, G. (2019). The role of micro-moments: A survey of habitual behavior change and recommender systems for energy saving. IEEE Systems Journal, 13(3), 3376-3387.

Kuznetsov, S., & Paulos, E. (2010, April). UpStream: motivating water conservation with low-cost water flow sensing and persuasive displays. In Proceedings of the SIGCHI Conference on Human Factors in Computing Systems (pp. 1851-1860).

He, H. A., Greenberg, S., & Huang, E. M. (2010, April). One size does not fit all: applying the transtheoretical model to energy feedback technology design. In Proceedings of the SIGCHI conference on human factors in computing systems (pp. 927-936).


- Other example papers about e-waste practices and barriers are here:
Vyas, Dhaval, and John Vines. "Making at the Margins: Making in an Under-resourced e-Waste Recycling Center." Proceedings of the ACM on Human-Computer Interaction 3.CSCW (2019): 1-23.

Dhir, A., Koshta, N., Goyal, R. K., Sakashita, M., & Almotairi, M. (2021). Behavioral reasoning theory (BRT) perspectives on E-waste recycling and management. Journal of Cleaner Production, 280, 124269.

---

### Official Review · Reviewer_zth4 · 2021-05-02
**Out of scope of Graphics Interface**

**Rating:** 3
**Confidence:** 4

**Review:**

The paper “I want to recycle batteries, but it’s inconvenient” presents a combined survey and interview study of the practices and challenges around battery recycling.
The authors survey 106, primarily North American subjects between 18 and 35 years old, and interview 12 of them. They identify a number of challenges with how participants lack knowledge of recycling options and motivation for reusing batteries still holding charge.

While the paper addresses a commendable topic, I find that it is out of scope for the Graphics Interface conference. There is simply little of the contribution that relates to HCI or computer graphics. The paper does touch upon a design suggestion for a battery tester and accompanying app. However, the design is very superficially discussed, loosely based on the empirical data, and not evaluated in any form.

Regarding the study, I miss a discussion of the representativeness of the recruited participants and the recruitment criteria used. It seems that the age group is focused around young adults and the participants primarily from the US. However, without knowing demographic details about the participants, it is not easy to assess, e.g., the cultural differences in recycling between countries or the differences between homeowners and tenants.

Overall, I cannot recommend accepting the paper for Graphcis Interface 2021.

---

### Official Review · Reviewer_BMcZ · 2021-05-03
**Unclear contribution**

**Rating:** 1
**Confidence:** 3

**Review:**

The paper presents the results from a survey and interview on battery reuse and recycling practices. Authors distilled the results in three opportunities for design and discuss design implications. Unfortunately, I find it hard to identify the contribution of the paper to the graphic interface or the hci community. The paper offers very broad and superficial design implications. The authors only provide a proof-of-concept interface of a mobile app to show battery information (DC1). It is not clear how the proposed design should "lower the barrier for the general public to test used batteries." This proof-of-concept has not been tested with users, and therefore we cannot really learn anything from it. It is also unclear how the other two design considerations should be taken into account for the design of interactive systems. The discussion, again, is too shallow.

---

### Meta-Review · Area_Chair_rJ8o · 2021-05-06

**Recommendation:** Reject
**Confidence:** 5

**Metareview:**

The reviewers all agree that the paper's contribution is too weak and out of the scope of the Graphics Interface conference.
One reviewer points out that most of the issues identified in the study should be addressed by city recycling infrastructure and not tech.
The reviewers also agree that the design contribution (in the form of a static mock-up) is too weak and not evaluated, so it is difficult to learn something from it.

Overall my recommendation is to reject the paper.

---

### Decision · Program_Chairs · 2021-05-08

Reject